# A New *Bacillus velezensis* Strain CML532 Improves Chicken Growth Performance and Reduces Intestinal *Clostridium perfringens* Colonization

**DOI:** 10.3390/microorganisms12040771

**Published:** 2024-04-11

**Authors:** A La Teng Zhu La, Qiu Wen, Yuxuan Xiao, Die Hu, Dan Liu, Yuming Guo, Yongfei Hu

**Affiliations:** State Key Laboratory of Animal Nutrition and Feeding, College of Animal Science and Technology, China Agricultural University, Beijing 100193, China; zhula8336@163.com (A.L.T.Z.L.); wenqiu814@163.com (Q.W.); xiaoyuxuan12@163.com (Y.X.); hudie19991109@163.com (D.H.); liud@cau.edu.cn (D.L.); guoyum@cau.edu.cn (Y.G.)

**Keywords:** *Bacillus velezensis*, probiotic, *Clostridium perfringens*, gut microbiota, growth performance

## Abstract

*Bacillus velezensis* has gained increasing recognition as a probiotic for improving animal growth performance and gut health. We identified six *B. velezensis* strains from sixty *Bacillus* isolates that were isolated from the cecal samples of fifteen different chicken breeds. We characterized the probiotic properties of these six *B. velezensis* strains. The effect of a selected strain (*B. velezensis* CML532) on chicken growth performance under normal feeding and *Clostridium perfringens* challenge conditions was also evaluated. The results revealed that the six *B. velezensis* strains differed in their probiotic properties, with strain CML532 exhibiting the highest bile salt and acid tolerance and high-yield enzyme and antibacterial activities. Genomic analyses showed that genes related to amino acid and carbohydrate metabolism, as well as genes related to starch and cellulose hydrolysis, were abundant in strain CML532. Dietary supplementation with strain CML532 promoted chicken growth, improved the gut barrier and absorption function, and modulated the gut microbiota. Under the *C*. *perfringens* challenge condition, strain CML532 alleviated intestinal damage, reduced ileal colonization of *C*. *perfringens*, and also improved chicken growth performance. Collectively, this study demonstrated that the newly isolated *B. velezensis* strain is a promising probiotic with beneficial effects on chicken growth performance and gut health.

## 1. Introduction

The gut microbiota is crucial for preserving gut health and influencing the overall performance of chickens by regulating nutrient digestion, the immune system, and gut function. The complex microbial interactions and metabolites that the members of the microbial community produce or that result from the conversion of host molecules or diet mediate these modulating effects [1]. Utilizing different feed additives, such as prebiotics and probiotics, may fortify and improve the animal gut microbiota composition and microbial interactions. Probiotics are considered the most promising feed ingredients for managing the gut microbiota. They can strengthen the gut microbiota, optimize animal performance, and enhance colonization resistance to intestinal pathogens. Probiotics can be single-strain or multiple-strain combinations of specific species, with the more common ones belonging to the genera *Bacillus*, *Lactobacillus*, *Saccharomyces*, *Enterococcus*, and *Bifidobacterium* [2,3,4,5,6]. Although a huge number of probiotics have been defined, the search for novel probiotic strains with fine properties remains of interest due to the broad usage of probiotics in animal husbandry.

Currently, there has been growing interest in selecting potential probiotic strains from the host’s indigenous intestinal microbiota based on the assumption that the indigenous microorganisms have a mutually beneficial interaction with the host, enabling them to better colonize the gastrointestinal tract. However, not all kinds of probiotics, regardless of origin, can survive well during the feed production process and when entering the chickens’ gastrointestinal tract [7], except for *Bacillus*-based probiotics. *Bacillus* probiotics have spore-forming capabilities [8], enabling them to withstand damage during feed manufacturing processes and endure the harsh circumstances in the gastrointestinal tract, including the presence of low pH and bile salt conditions. In addition, *Bacillus* spp. is also recognized for its rapid growth rate, extensive production of digestive enzymes, and capacity to outcompete certain pathogenic bacteria [2]. Therefore, it is promising to isolate *Bacillus* strains as animal probiotics for improving animal growth performance and gut health, especially in the gastrointestinal tract of target animal species.

Due to *B. velezensis*’s special ability to inhibit plant diseases, it is widely used as a biological control agent in the agricultural field [9]. Different strains of this species have been reported to suppress the growth of microbial pathogens, such as bacteria and fungi, by producing secondary metabolites [10]. The advantageous function of *B. velezensis* as a feed additive has also been proven in poultry. *B. velezensis* ZBG17 isolated from soil samples enhanced feed utilization efficiency and heightened the humoral immune response in broiler chickens [11]. Similarly, *B. velezensis* LB-Y-1 obtained from the digestive tracts of free-ranging animals promoted broiler chickens’ growth performance and the mineralization of tibias by enhancing the activities of intestinal digestive enzymes, increasing phosphorus metabolism and utilization, and altering the intestinal microbiota [12]. *B. velezensis* isolated from piglet manure has been shown to improve the laying hens’ egg production and quality, as well as plasma biochemical indicators [13]. These findings indicated that *B. velezensis* is a beneficial dietary additive for enhancing chicken growth and health, and it has the potential to be a novel probiotic in poultry production. However, the majority of the isolated *B. velezensis* strains that promote growth performance in chickens originate from other animals than the native host, and a comparison of the probiotic properties of different *B. velezensis* strains is limited. Additionally, as a known bacterial species that has broad antimicrobial activity, whether *B. velezensis* can inhibit a chicken bacterial infection such as *Clostridium perfringens* that causes necrotic enteritis is not fully understood.

In the current study, we isolated and identified autochthonous *B. velezensis* from locally reared Chinese local-breed chickens and assessed the capabilities of six *B. velezensis* strains in vitro. We sequenced the complete genome of the selected strain *B. velezensis* CML532, analyzed its gene contents, and evaluated its safety regarding the existence of virulence genes and antibiotic resistance genes. We finally investigated the effect of *B. velezensis* CML532 on growth performance and gut health under normal feeding and *C*. *perfringens* challenge conditions in broiler chickens.

## 2. Materials and Methods

### 2.1. Sample Collection, Isolation, and Identification of Bacillus Strains

Fifteen different local breeds of chickens were slaughtered in various regions of China. Broiler cecum was isolated and preserved at −20 °C. *Bacillus* strain was isolated using the method previously described with minor modifications [12]. The cecal contents were homogenized with phosphate-buffered saline at 1:9 (*w*/*v*) ratio, and the compound was then heated in an 80 °C water bath for 10 min. After cooling to room temperature, the sample was diluted with sterile phosphate-buffered saline until it reached the appropriate concentration. Then, 100 μL was inoculated in triplicate by surface spreading on Luria–Bertani (LB, Beijing Aoboxing Biotechnology Co., Ltd., Beijing, China) liquid medium and cultured at 37 °C for 24 h. Single colonies suspected of being *Bacillus* were selected from the LB plate and cultured for 24 h at 37 °C. The purified single colonies were then inoculated in LB medium and incubated on a shaking table (Pei Ying Experimental Equipment Co., Ltd., Suzhou, China) at 37 °C and 200 rpm for 12 to 24 h. The target bacterium was further identified through the analysis of its 16S rRNA gene sequence. The genomic DNA of the bacterium was collected from the screened individuals with a DNA isolation kit (Jiangsu Cowin Biotech Co., Ltd., Beijing, China). The 16S rRNA primers used were 27F (5′-AGAGTTTGATCCTGGCTCAG-3′) and 1492R (5′-GGTTACCTTGTTACGACTT-3′). The 16S rRNA gene sequences were used for bacterial species identification by conducting a BLAST search in the EZBioCloud database (https://www.ezbiocloud.net/) (accessed on 23 January 2022). The *Bacillus* isolates were evaluated for their probiotic properties, including tolerance to acid, bile salts, and high temperatures; the ability to secrete proteases, amylases, and cellulases; and the inhibitory effects on *Escherichia coli*, *Salmonella typhimurium*, and *C. perfringens*. More details on the probiotic properties are included in Appendix A.

### 2.2. Complete Genome Sequencing and Analysis of Bacillus velezensis CML532

A single colony of *B. velezensis* CML532 was cultured in broth medium (Beijing Aoboxing Biotechnology Co., Ltd., Beijing, China) at 200 rpm and 37 °C for 12 h. Genomic DNA was extracted and sequenced on the PacBio high-fidelity sequencing platform (Beijing Biomarker Technologies Co., Ltd., Beijing, China). Fastp and FastQC were utilized for raw reads filtering and quality control, respectively. Clean reads were assembled into contiguous sequences utilizing Hifiasm [14], and their quality was evaluated using Pilon (v1.22) [15]. The genome was annotated, including coding sequences (CDSs), rRNA, tRNA, etc., by using Prokka [16], and a genome map was generated based on the annotated results using Circos (v0.66) [17]. Functional annotation was performed by using the KEGG [18], GO [19], eggnog [20], ResFinder [21], and VFDB [22] databases. The CAZy database was utilized for further analysis of CAZyme (carbohydrate-active enzyme) [23], and the transporters were annotated based on the Transporter Classification Database (TCDB) using the BLASTing software (https://www.tcdb.org/progs/blast.php) (accessed on 16 March 2022) [24].

### 2.3. Fermentation and Preparation of Bacterial Powder

*B. velezensis* CML532 was cultured in sporulation medium (10 g glucose, 20 g (NH_4_)_2_·SO_4_, 0.5 g MgSO_4_·7H_2_O, 0.5 g MnSO_4_, 1 g K_2_HPO_4_, 2 g KH_2_PO_4_, and 3 g CaCO_3_ per liter of distilled water) at 37 °C for 24 h with 200 rpm. The above reagents were purchased from Sinopharm Chemical Reagent Co., Ltd. (Beijing, China). After incubation, the culture was centrifuged at 4 °C for 30 min with 4000 rpm, and defatted milk powder (Beijing Coolaibo Technology Co., Ltd., Beijing, China) and 5% trehalose (Shanghai Aladdin Biochemical Technology Co., Ltd., Shanghai, China) were added as protective agents. The compound was freeze-dried and ground into a powder, which was stored at −20 °C.

### 2.4. Dietary Supplementation of B. velezensis CML532 in Chickens under Normal Feeding Conditions

The experiment was conducted in conformity to the China Agricultural University Animal Care and Use Committee (AW30112202-1-1, Beijing, China). One-day-old Arbor Acres chicks were randomly assigned to two groups: control (CON) and *B. velezensis* CML532 (1 × 10^9^ CFU/kg diet). Each treatment included 6 replicates with 10 birds per replicate (10 birds/cages). All chickens were fed ad libitum water and were fed without antibiotics. The basal diet’s composition and nutrient content are presented in Appendix A. The bodyweight (BW) and feed intake (FI) per cage were measured at days 22 and 42. One broiler was randomly selected from each replicate group, and blood, jejunum digesta and tissue, and cecal digesta and tissue were collected at day 42. Serum was obtained by centrifuging the blood at 4 °C and 3000× *g* for 10 min and then kept at −20 °C. Intestinal digests and tissues were quick-frozen in liquid nitrogen and then transferred to −80 °C for DNA and RNA extraction and enzyme activity detection or placed in a 4% paraformaldehyde solution (Wuhan Servicebio Technology Co., Ltd., Wuhan, China) for histological evaluation.

### 2.5. Dietary Supplementation of B. velezensis CML532 under the C. perfringens Challenge Condition in Chickens

Three groups were assigned in this experiment as follows: negative control [(CON), no CML532 addition and no challenge with *C. perfringens*], *C. perfringens*-challenged control [(*Cp*), no CML532 addition but challenged with *C. perfringens*], and *C. perfringens*-challenged + *B. velezensis* supplementation [(CB), *B. velezensis* CML532 (1 × 10^9^ CFU/kg diet) and challenged with *C. perfringens*]. Each treatment included 7 replicates with 12 one-day-old Arbor Acres chicks per replicate (12 birds/cages). From days 14 to 20, the chickens in the CON group were orally administered 1 mL of fluid thioglycolate medium (FTG, Beijing Aoboxing Biotechnology Co., Ltd., Beijing, China). At the same time, the chickens in the remaining groups were orally administered the same amount of FTG culture containing 1 × 10^9^ CFU of *C. perfringens* (CVCC37, China General Microbiological Culture Collection Center, Beijing, China). All the chickens had free access to water and feed that did not contain antibiotics. The composition and nutrient content of the basal diet are presented in Appendix A. Individual BW and FI by cage were measured at days 13, 20, and 40. One broiler was randomly selected from each replicate group, and jejunum tissue, ileum digesta, and tissue were collected at day 20. Intestinal digests and tissues were quick-frozen in liquid nitrogen and then transferred to −80 °C for DNA and RNA extraction.

### 2.6. Bird Management

A three-layered chicken coop was utilized, and the birds were reared at a stocking density (ranging was 14.3 birds/m^2^). Temperature was progressively reduced from 33 °C on day 0 and ending at 21 °C by the end of the trial. The birds received immunizations in accordance with standard commercial practices. The indices of humidity, lighting, temperature, and hygiene within the chicken house were maintained in accordance with the hygienic standards for broilers as stipulated in the regulation GB 14925-1994.

### 2.7. Organ Index

The chickens were sacrificed and measured at the end of the experiment. The liver, spleen, bursa of Fabricius, thymus, and abdominal fat were collected to calculate the visceral indices.

### 2.8. Serum Parameter Assay

To evaluate the prooxidant–antioxidant balance in serum, we determined CAT (catalase), GSH-Px (glutathione peroxidase), SOD (superoxide dismutase), and GST (glutathione S-transferase) activities. All antioxidant-related indices were quantified using commercial kits purchased from Nanjing Jiancheng Bioengineering Institute (Nanjing, China).

### 2.9. Intestinal Morphological Structure

Jejunum tissues fixed with paraformaldehyde were dehydrated using a series of ethanol concentrations embedded in paraffin and then cut into 2 μm thick cross-sections using a microtome and stained with H&E [25]. Photomicrographs were taken at 200× magnification using a Leica microscope (DM750, Leica Microsystems Co., Ltd., Shanghai, China). The crypt depth and villus height of jejunum samples were measured from all complete, vertically oriented villi in each slide.

### 2.10. Intestinal Lesion Score

The intestinal samples from chickens were scored blindly based on Dahiya et al. [26], with minor adjustments. More details on the intestinal lesion score are included in Appendix A.

### 2.11. Intestinal Disaccharidase Activity Measurement

The mucosal scrapings collected from the jejunum were homogenized (1:9 *w*/*v*) in physiological saline and then centrifuged at 4 °C for 15 min with 3000× *g*. The protein content and the activities of maltase, Na^+^/K^+^-ATPase, alkaline phosphatase (ALP), and sucrase were evaluated using commercial kits provided by Nanjing Jiancheng Bioengineering Institute (Nanjing, China). Enzyme activity in the jejunum mucosa was normalized to the corresponding protein concentration and expressed as U/mL of protein.

### 2.12. Total RNA Isolation and Quantitative Real-Time PCR

RNA was extracted from the intestinal tissue using RNAiso Plus (Beijing Tsingke Biotech Co., Ltd., Beijing, China). RNA content and purity were measured using a NanoDrop 2000 spectrophotometer (Thermo Scientific, Waltham, MA, USA). Then, 1 µg of RNA was reverse-transcribed to cDNA using a BeyoRT™ II First Strand cDNA Synthesis Kit with a gDNA Eraser (Beyotime Biotechnology, Shanghai, China). One-step real-time PCR was performed with the BeyoFastTM SYBR Green qPCR Mix (Beyotime Biotechnology, Shanghai, China) on an ABI 7500 fluorescence quantitative PCR instrument (Thermo Scientific, Waltham, MA, USA). The primers were synthesized by Invitrogen (Shanghai, China), and the sequences are presented in Appendix A. Relative gene expression was calculated using the 2^−ΔΔCt^ method with *β-actin* used for normalization.

### 2.13. DNA Extraction and Quantification, Amplicon Sequencing, and Data Processing

The QIAamp DNA Stool Mini Kit (Qiagen Inc., Valencia, CA, USA) was utilized to extract bacterial DNA from the cecal contents of chickens. The V3–V4 regions of the 16S rRNA were amplified from microbial genomic DNA (341F: 5′-CCTACGGGNBGCASCAG-3′; 805R: 5′-GACTACNVGGGTATCTAATCC-3′) using a polymerase chain reaction, as described previously [27]. The Illumina HiSeq PE250 platform (Illumina, Inc., San Diego, CA, USA) was utilized to perform the paired-end sequencing. The raw fastq files underwent quality filtering, denoising, merging, and amplicon sequence variant assessment using DADA2 [28]. Taxonomic assignment was conducted using UNITE (v8) [29] and SILVA (v132) [30]. The α-diversity and principal coordinate analysis (PCoA) were computed using vegan (v2.5-7) according to the Bray–Curtis distance. The differential abundance of taxa was determined using LEfSe (linear discriminant analysis effect size) with the parameter “LDA ≥ 2”. Co-occurrence networks were constructed according to the Spearman correlations, and the corresponding *p*-values for multiplicity were adjusted using the Benjamini–Hochberg correction. Using the R package iGraph [31], we defined a co-occurrence event as a correlation with a Spearman’s correlation coefficient > |0.7| and *p*-value < 0.05. Cytoscape 3.5.1 explored and visualized the co-occurrence network [32]. To identify network modular structures, we used modularity (M) as a threshold of M > 0.4 [33].

### 2.14. DNA Extraction and Enumeration of Intestinal C. perfringens

The quantity of *C. perfringens* in the ileal content was determined by the absolute real-time quantitative PCR method outlined by Du and Guo [34]. Briefly, bacterial genomic DNA was extracted using the stool genomic DNA extraction kit (Jiangsu CoWin Biotech Co., Ltd., Taizhou, China). A target DNA concentration was prepared by PCR amplification using DNA isolated from *C. perfringens*. The primers (F: AAAGATGGCATCATCATTCAAC; R: TACCGTCATTATCTTCCCCAAA) were designed based on the 16S rDNA sequence of *C. perfringens*. PCR products were purified and recovered by the Gel Extraction Kit (Jiangsu CoWin Biotech Co., Ltd., Taizhou, China). The purified products were connected with the pMD19-T vector using a TA cloning kit (Jiangsu CoWin Biotech Co., Ltd., Taizhou, Taizhou, China) and transformed into *E. coli* DH5α (Tiangen Biochemical Technology Co., Ltd., Beijing, China) to generate standard plasmids. The concentration of the purified plasmid with the insert was determined, and the number of target gene copies was computed using the subsequent formula:(1)$$DNA\left(copy\right)=\frac{6.02\times 10^{23}\left(copy/mol\right)\times DNA\;amount\left(g\right)}{DNA\;length\left(bp\right)\times 660\left(g/mol/bp\right)}$$

The genomic DNA extracted from ileal content was used as a template for real-time quantitative PCR, and plasmid DNA with a 10-fold continuous dilution was added to the PCR plate. The results are expressed as log_10_ copies of the gene per gram of content.

### 2.15. Statistical Analysis

Statistical analysis was performed using SAS Statistics 9.4 software. All data were tested for normality. If the data were not normally distributed, non-parametric tests were used for their analyses. The data from the normal feeding trial, including growth performance, intestinal morphology, organ index, enzyme activities, and the gene expression levels collected for quantitative parameters, were analyzed using Student’s *t* test. Data related to the growth performance, intestinal lesion score, the quantity of *C. perfringens*, organ index, and the gene expression levels were analyzed in a completely randomized design using the GLM procedures of SAS Statistics 9.4 software. Wilcoxon rank sum test was used for Bray–Curtis distance. *p* < 0.05 represented statistically significance; 0.05 ≤ *p* ≤ 0.10 represented a tendency toward significance.

## 3. Results

### 3.1. Isolation and Characterization of B. velezensis Strains from Chinese Local-Breed Chickens

To find new *B. velezensis* strains from autochthonous chicken gut microbes, we isolated *Bacillus* strains from the cecal samples of 15 locally reared Chinese local-breed chickens (Appendix A). A total of 60 strains, representing 20 different bacterial species, were isolated and identified. *B*. *altitudinis* was the most prevalent *Bacillus* species in chickens (9/60 isolates; 15.0%), followed by *B. paralicheniformis* (8/60 isolates; 13.3%), *B. tequilensis* (8/60 isolates; 13.3%), *B. siamensis* (7/60 isolates; 11.7%), and *B. velezensis* (6/60 isolates; 10.0%). The six *B. velezensis* strains were isolated from different breeds of chickens, including Shanzhongxian rooster (CML532), Shanzhongxian hen (CML526), Bairi (CML537), Shiqi (CML539), Luhua (CML542), and Langya (CML546).

We then evaluated the tolerance of the six strains of *B. velezensis* to low pH, different bile salt concentrations, and high temperatures. These characteristics are preconditions for a specific probiotic that can be used as a chicken feed additive. At pH 3, after 2 h of exposure, the survival rate of all the strains was greater than 58%. Four strains exhibited a satisfactory survival rate of over 55% after being exposed to pH 2 for 2 h (Figure 1A), among which *B. velezensis* CML532 is the highest tolerant strain with a survival rate of 108.2%. We found that five isolates exhibited resistance to both 0.3% and 0.4% bile salts; *B. velezensis* CML532 displayed the highest tolerance compared to the others under the 0.3% bile salt condition, while both *B. velezensis* CML532 and CML526 exhibited relatively better performance under the 0.4% bile salt condition (Figure 1B). To evaluate the survivability of *B. velezensis* strains at high temperatures, we measured their survival rates after exposure to temperatures of 85 °C and 90 °C for 10 min. The results revealed that three strains (CML546, CML532, and CML526) exhibited a high survival rate of over 100.0% after being exposed to 85 °C for 10 min (Figure 1C). Taking into account all the tolerance characteristics of the six *B. velezensis* strains, we selected *B. velezensis* CML532 for further investigation. CML532 was capable of producing different enzymes, including amylase, protease, and cellulase (Figure 1D), and inhibited the growth of the common chicken microbial pathogens *E. coli*, *S. typhimurium*, and *C. perfringens* (Figure 1E). CML532 exhibits a relatively stronger inhibitory effect against *C. perfringens* compared with *E. coli* and *S. typhimurium* (Figure 1E). Taken together, high tolerance to bile salt and acid, a high yield of enzymes, and antibacterial activities warrant the application of *B. velezensis* CML532 as a potential probiotic in chicken feeding.

### 3.2. Genomic Analysis of B. velezensis CML532

The *B. velezensis* CML532 genome was sequenced, and a complete genome of 4,003,723 bp with a G + C content of 46.53% was obtained. The whole genome contained 4125 predicted protein-coding genes with an average gene length of 857 bp. The *B. velezensis* CML532 genome sequence contains 27 rRNA- and 86 tRNA-coding genes. Protein functional annotation analysis using the COG database revealed that [G] carbohydrate transport and metabolism, [E] amino acid transport and metabolism, and [K] transcription were the top three categories containing the highest number of known genes (Figure 2A). The KEGG annotation showed that the functional genes were more related to the ABC transporters, biosynthesis of amino acids, carbon metabolism pathways, and two-component system (Figure 2B). GO functional analysis indicated that genes belonging to metabolic process, catalytic activity, and membrane accounted for the highest proportion in the categories of biological processes, molecular functions, and cellular component ontologies (Figure 2C).

We then annotated the genome of *B. velezensis* CML532 using the CAZyme and TCDB databases. The CAZyme annotation results revealed that strain CML532 possessed 3 polysaccharide lyase genes, 7 auxiliary activity genes, 31 carbohydrate esterase genes, 37 carbohydrate-binding module genes, 40 glycosyltransferase genes, and 47 glycoside hydrolase genes (Figure 3A). For TCDB analysis, 1078 genes were categorized into 7 transporter classification systems, including incompletely characterized transport systems, channels/pores, group translocators, primary active transporters, transmembrane electron carriers, accessory factors involved in transport, and electrochemical potential-driven transporters (Figure 3B).

To evaluate the safety of *B. velezensis* CML532 at the genomic level, we performed genome annotation to identify virulence and antibiotic resistance genes using the VFDB and ResFinder databases. The results showed that *B. velezensis* CML532 did not contain any known virulence genes but possessed the antibiotic resistance genes *cfr*(B) and *tet*(L), displaying 89.09% and 86.86% similarities with their counterparts in the database (Appendix A). However, no mobile genetic elements were found in the 10 kb regions flanking these two genes as annotated by the MobileElementFinder tool, suggesting these two genes are intrinsic resistance genes that cannot move through horizontal gene transfer. Further, the results of antibiotic susceptibility tests indicated that *B. velezensis* CML532 was susceptible to 29 antibiotics tested and exhibited resistance only to tetracycline (Table 1), probably due to the existence of the active intrinsic *tet*(L) gene [35].

### 3.3. B. velezensis CML532 Improves the Chicken’s Growth Performance and Gut Function

In order to investigate the impacts of *B. velezensis* CML532 on growth performance, 1-day-old chicks (two groups; 6 replicates/group × 10 chicks/replicate) were given a diet with or without *B. velezensis* CML532 for 42 days. Compared with the control group, strain CML532 significantly increased BW at d 42 (*p* < 0.05, Figure 4A) and improved the ADG and ADFI at d 22–42 and d 1–42 (*p* < 0.05, Figure 4B,C). However, no obvious changes (*p* > 0.05) in F/G ratio were observed (Figure 4D). CML532 supplementation obviously increased (*p* < 0.05, Figure 4E–G) the villus length and villus/crypt ratio of jejunum relative to the CON group; however, the crypt depth was similar between the two groups (*p* > 0.05). We observed a significant decrease in the liver index of chickens fed with CML532 (*p* < 0.05, Figure 4H). We also found a significant increase in the ALP and maltase activities of chickens fed with CML532 (*p* < 0.05, Figure 4I). The data for the gene expression of *occludin*, *claudin*-2, *claudin*-3, and *ZO*-1 (*p* < 0.05) were all profoundly improved at day 42 (Figure 4J). Of note, the gene expression of *PEPT* and *TIRI* was significantly improved by CML532 supplementation (*p* < 0.05), and the gene expressions of *SGLT1*, *GLUT1*, and *TIR3* tended to be higher at d 42 (*p* < 0.1, Figure 4K). Additionally, the gene expression of the cytokine IL-8 was upregulated (*p* < 0.1) in the cecal tonsils of the CML532 group at d 42 (Figure 4L), indicating that in-feed CML532 regulated chicken immunity. *B. velezensis* CML532 supplementation promoted increases (*p* < 0.05) in SOD activity in the serum (Figure 4M). In summary, dietary supplementation with *B. velezensis* CML532 promoted chicken growth and improved digestion and absorption function, intestinal morphological structure, and barrier function.

### 3.4. B. velezensis CML532 Alters the Chicken Cecal Microbial Community Structure

In order to investigate the impact of *B. velezensis* CML532 on the gut bacterial community in chickens, we conducted 16S rRNA gene amplicon sequencing of the cecal content. We identified 10 bacterial phyla having a relative abundance of more than 0.1%, with Firmicutes, Bacteroidota, Proteobacteria, Verrucomicrobia, and Cyanobacteria being the top five most abundant phyla in the gut microbiota (Appendix A). Unknown *Lachnospiraceae* was the most abundant genus in the cecal microbiota, followed by *Faecalibacterium*, *Streptococcus*, *Ruminococcaceae* UCG-014, and *Ruminococcaceae* UCG-005 (Figure 5A). There were no differences in the α-diversity between the control and CML532 groups (Appendix A), while the two groups were clearly separated in the β-diversity analysis based on Bray–Curtis distance (Figure 5B,C).

We then investigated which microbes in the gut microbiota are mostly impacted by strain CML532. The LEfSe results demonstrated that 16 genera exhibited differential abundance between the control and CML532 groups. The LEfSe analysis revealed that *Blautia*, *CHKCI001*, *Ruminiclostridium*-5, *Delftia*, *Sellimonas*, *Lachnospira*, *uncultured Erysipelotrichaceae*, *Defluviitaleaceae* UCG-011, *Coprobacter*, *Harryflintia*, and *Rothia* were significantly enriched in the CML532 groups, while *Bilophila*, uncultured *Veillonellaceae*, *Haemophilus*, and uncultured *rumen bacterium* were more abundant in the control groups (Figure 5D). The *Streptococcus* and unknown *Lachnospiraceae* were identified as the most prevalent and dominant core genus in the control and CML532 chickens, respectively, while *Bacteroides*, *Negativibacillus*, and *Butyricicoccus* were found to be the dominant core members specific to the CML532 chickens (Figure 5E). We next constructed the gut microbial co-occurrence network in the control and CML532 groups (Figure 5F,G). In total, the empirical networks included 200 nodes with 1081 edges for the control network and 202 nodes with 1066 edges for the CML532 network (Appendix A). The topological characteristics of the two networks were greater than those in the equivalent randomized networks, indicating that the constructed networks exhibited typical hierarchical, small-world, and modular properties (Appendix A). Both networks contained modules with M ≥ 0.54 (Appendix A), and the taxa with the highest modularity were *Eubacterium brachy* in the CML532 network and *Lachnospira* in the CON network (Figure 5F,G). Additionally, we found that the microbial taxa with higher degrees in the modules of the CML532 and CON networks were very different. These differences in the co-occurrence networks demonstrated that CML532 considerably altered the gut microbial interactions, which may have a profound impact on the gut microbial functions in chickens.

### 3.5. B. velezensis CML532 Suppresses Intestinal C. perfringens Colonization and Attenuates Intestinal Mucosal Injury

To determine the impact of *B. velezensis* CML532 on intestinal damage in broilers caused by *C. perfringens*, 1-day-old chicks (three groups; 7 replicates/group × 12 chicks/replicate) were orally challenged with or without *C. perfringens* from days 14 to 20 of age. The BW in the CB group was greater than that in the *Cp* and CON groups at d 13 and d 40 (*p* < 0.05, Figure 6A); the ADG was higher in the CB group than in the *Cp* and CON groups on d 1–13, d 21–40, and d 1–40 (*p* < 0.05, Figure 6B); the ADFI was higher in the CB group than in the CON group on d 1–13, d 21–40, and d 1–40 (*p* < 0.05, Figure 6C); the F/G ratio in the CB group was significantly lower than that in the *Cp* and CON groups on d 1–13 (*p* < 0.05, Figure 6D). After *C. perfringens* infection, the chicken jejunum was severely damaged, as reflected by the macroscopic lesions characterized by thin-walled and fragile intestines with tiny hemorrhagic spots, while the lesion score was lower in the challenged chickens fed with CML532 (*p* < 0.05, Figure 6E). The challenge of *C. perfringens* led to an obvious increase in the population of ileal *C. perfringens* (*p* < 0.05) that was significantly decreased after CML532 treatment (*p* < 0.05, Figure 6F). The bursa of Fabricius index was remarkable higher (*p* < 0.05) in the *Cp* birds than in the CB and CON groups (Figure 6G). The gene expression levels of IL-1β and INF-γ were significantly lower in the *Cp* birds than those in the CB and CON birds, and the expression of *mucin*-2 and *occludin* was also higher (*p* < 0.05) in the *Cp* birds than those in the jejunum of the CB and CON birds (Figure 6H). The relative expression levels of INF-γ in the ileum of CB birds were significantly lower than those of the control birds (*p* < 0.05, Figure 6I). Additionally, the *Cp* birds significantly increased the expression of *mucin*-2 and *occludin* compared with the CB birds in the ileum (*p* < 0.05, Figure 6I). Overall, *B. velezensis* CML532 alleviated intestinal damage, modulated the immune response, reduced ileal colonization of *C. perfringens*, and also improved chicken growth performance under *C. perfringens* challenge conditions.

## 4. Discussion

Our results showed that *B. velezensis* CML532 exhibited high tolerance to bile salt and acid circumstances, produced high levels of extracellular enzymes, and inhibited common chicken gut pathogens. The reason for the high tolerance to bile salt and acid of *B. velezensis* CML532 could be attributed to the resistant nature of *Bacillus* spores. Although chickens naturally produce these digestive enzymes, the additional production of them from gut probiotics further helps the host digest carbohydrates and proteins, thereby increasing nutrient absorption [36]. In addition, *B. velezensis* CML532 produces a high level of cellulase, a non-starch polysaccharide enzyme, which may increase the surface-to-volume ratio of feed particles and improve the efficiency of energy conversion [2]. *B. velezensis* CML352 displayed an inhibitory effect on *S. typhimurium*, *C. perfringens*, and *E. coli*, with the highest antimicrobial activity being against *C. perfringens*. This pathogen is known as the major causative agent that causes necrotic enteritis, an economically important disease in chicken production [37]. Collectively, these probiotic features suggest the potential of using *B. velezensis* CML532 as a feed additive to improve chicken gut health and growth performance.

Using third-generation sequencing technology (PacBio sequencing), we sequenced the complete genome of *B. velezensis* CML532. Genome analysis revealed that strain CML532 possesses a large number of genes related to the synthesis and transport of amino acids and carbohydrates, indicating that *B. velezensis* CML532 is highly active in the processing and transport of nutrients. Additionally, GO annotation results showed that functional genes were most enriched in the catalytic activity category (1886 genes), suggesting that *B. velezensis* CML532 has a strong physiological metabolism and a high potential to produce complex and diverse metabolites. Moreover, the diverse composition of CAZyme-encoding genes in the genome may contribute to the strong cellulolytic properties of *B. velezensis* CML532. Similar results regarding the versatile functional activities of *B. velezensis* have also been revealed in other strains. *B. velezensis* LB-Y-1 was found to produce cellulase, protease, lipolytic, amylolytic, and phytase [12]; *B. velezensis* P11 isolated from the Brazilian Amazon basin had extraordinary keratinolytic, proteolytic, and dehairing activities [38]; and diverse strains of *B. velezensis* were verified to improve host disease resistance by producing secondary metabolites, such as fengicins, amylocyclicin, surfactins, and bacillomycin, among others [39]. Our genomic analysis results also demonstrated that *B. velezensis* CML532 did not carry known virulence genes or mobile antibiotic resistance genes, indicating that there are no safety concerns regarding virulence and antibiotic resistance genes transfer when using *B. velezensis* CML532 as a feed additive.

We demonstrated that the inclusion of *B. velezensis* CML532 into the feed significantly improved the growth performance of chickens. It is well known that probiotics can enhance growth performance by promoting the establishment of a healthy gut environment [40]. Consistent with this, the enhanced growth performance of broilers after treatment with *B. velezensis* CML532 was attributed to an increase in gut function, as reflected by gut morphology, associated enzymatic activity, and barrier function. *B. velezensis* CML532 also enhanced the digestion and absorption capacities of chicken intestines, as demonstrated by the increased jejunum villi length, villi/crypt ratio, and brush border enzymes, which could promote an increase in the intestine absorptive surface and create an advantageous environment for nutrient uptake in the intestine [41]. Additionally, *B. velezensis* CML532 enhanced the activity of ALP, an enzyme that was known to promote gut barrier function mainly through upregulating intestinal tight junction proteins in chickens [42].

The chicken gut microbiota and its metabolic products have a role in enhancing chicken growth and maintaining gut health [43]. *Bacillus* spp. has been demonstrated to alter the chicken gut microbiota, reduce competition for nutrients between microbes and the host, and improve gastrointestinal health [44]. Previous studies have shown that supplementing with *B. velezensis* not only maintains the balance of gut microbiota in the host but also increases the colonization of beneficial bacteria and reduces the colonization of harmful bacteria [12,45]. Similarly, our research revealed that *B. velezensis* CML532 significantly changed the composition of the chicken cecal microbiota by enriching the beneficial microbe’s abundance. For example, CML532 remarkably increased the abundance of *Blautia*, a genus of anaerobic bacteria with probiotic characteristics widely found in the gut [46]. A recent finding showed that a strain of *Blautia* product effectively suppressed gut inflammatory responses and maintained the intestinal barrier [47]. Therefore, in our study, the improvement in intestinal mucosal barrier function after *B. velezensis* CML532 intervention could be to a certain degree attributed to the significantly increased abundance of *Blautia*. We also found that several beneficial microbial taxa, such as *Pseudoflavonifractor*, *Eubacterium brachy*, and *Eubacterium hallii*, in the CML532 network had frequent interactions (higher degree) with other gut microbes. Overall, we propose that dietary *B. velezensis* CML532 alters the gut microbiota, improves the abundance of beneficial bacteria, and thereby enhances microbial interaction stability in the chicken.

As *B. velezensis* CML532 displayed an inhibitory effect on *C. perfringens* in our in vitro studies, we supposed that dietary *B. velezensis* CML532 could reduce infection and improve gut health in chickens challenged with *C. perfringens*. We showed that the *C. perfringens* challenge condition induced macroscopic pathological changes in the gut, such as hyperemia, hemorrhage, and a thin-walled and brittle gut. In addition, pathogen challenges have led to an extended bursa of Fabricius and a high rate of *C. perfringens* colonization in the ileum. The findings were in line with previous studies [48], which demonstrated that *C. perfringens* increased the bursa of Fabricius index. The immune organ index reliably indicates the level of immunoreactivity in broiler chickens. The spleen, bursa of Fabricius, and thymus play crucial roles as immune organs in broilers, with the bursa of Fabricius being an avian-specific immune organ where B lymphocytes begin to develop and acquire diversity in the antibody repertoire [49]. The changes in the bursa of Fabricius in our results demonstrate that *C. perfringens* challenges disrupted immune homeostasis.

The disruption of the intestinal mucosal barrier by intestinal pathogens damages intestinal immune tolerance, causes systemic inflammatory responses, exacerbates systemic infections, and damages the host organism [50]. Mucin-2 is the main part of mucin, which can effectively attach to various pathogens and suppress external bacteria’s access to epithelial cells [51]. The expression of *mucin*-2 is inhibited in infected broiler chickens [52]. However, *C. perfringens* infection increased the ileum relative expression level of *mucin*-2 in the chickens, suggesting that the pathogen-challenged broilers are able to eliminate the pathogens from the epithelial surface by promoting mucin expression [53,54]. Similar to the results of the expression of *mucin*-2, there was an obvious increase in infected *C. perfringens* chickens but a reduction after *B. velezensis* CML532. Consistent with this, we found the *C. perfringens* infection upregulated the relative expression levels of *occludin.* Our results, together with previous findings, suggest that upregulating the gene expression levels associated with barrier function in infected chickens may be one of the host defense mechanisms to reduce the passage of *C. perfringens* through the epithelial layers and minimize intestinal damage. Overall, we demonstrated that dietary *B. velezensis* CML532 reduces the intestinal lesion and ileal colonization of *C. perfringens*, improves the gut barrier, regulates the host’s immune systems, and thus promotes the chicken’s gut health and growth performance.

## 5. Conclusions

In conclusion, we isolated and illustrated the probiotic properties of six *B. velezensis* strains from Chinese local-breed chickens. We determined the complete genome sequence of a new strain, *B. velezensis* CML532, and demonstrated its beneficial effects on modulating growth performance, gut microbiota, intestinal morphology, intestinal digestion and absorption function, and immunity in chickens. We also revealed that *B. velezensis* CML532 ameliorated *C. perfringens*-induced intestinal damage in broilers by reducing intestinal lesions and *C. perfringens* ileal colonization. These findings highlight the promising role of *B. velezensis* as a probiotic for improving chicken growth performance and gut health in poultry production.

## Figures and Tables

**Figure 1 microorganisms-12-00771-f001:**
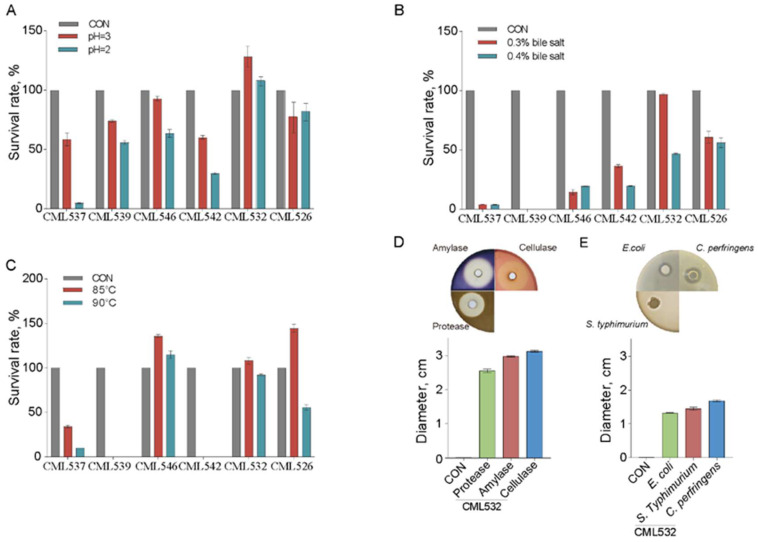
Probiotic properties of *B. velezensis* strains. Survival rate of different *B. velezensis* strains under (**A**) pH 2 and 3, (**B**) 0.3% and 0.4% bile salt, and (**C**) 85 °C and 90 °C. (**D**) Amylase, protease, and cellulase production capacity of *B. velezensis* CML532 measured by the zone diameter. (**E**) The antagonistic activity of *B. velezensis* CML532 against common chicken pathogens, namely *E*. *coli*, *S*. *typhimurium*, and *C*. *perfringens*. Data are expressed as the mean ± SEM (*n* = 3).

**Figure 2 microorganisms-12-00771-f002:**
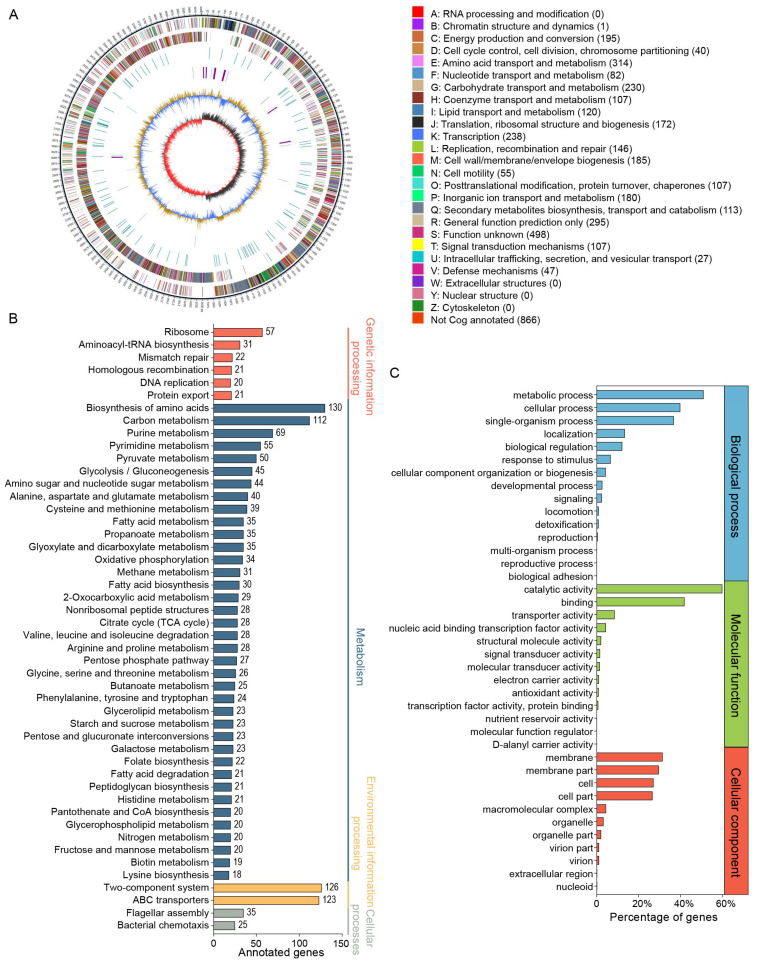
COG, KEGG, and GO functional annotations of predicted genes in *B. velezensis* CML532. (**A**) The genome map of *B. velezensis* CML532. (**B**) KEGG pathway and (**C**) GO functional annotation results.

**Figure 3 microorganisms-12-00771-f003:**
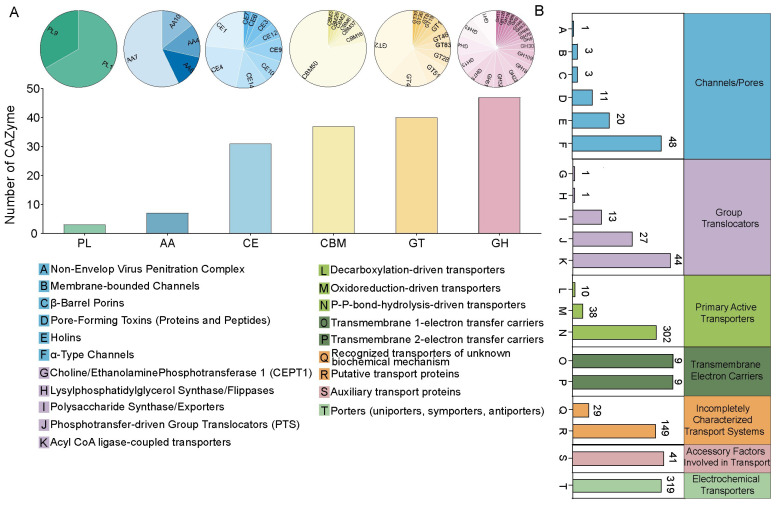
(**A**) Distribution of CAZymes in *B. velezensis* CML532. (**B**) Transporter classification and functional predictions of putative transport proteins from *B. velezensis* CML532.

**Figure 4 microorganisms-12-00771-f004:**
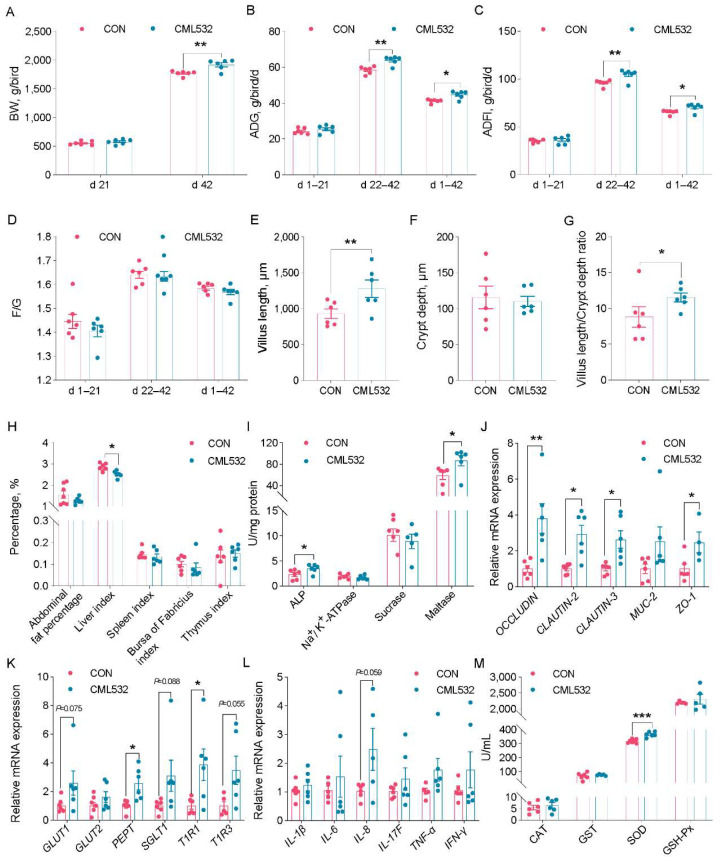
*B. velezensis* CML532 increases the growth performance and gut function in broiler chickens. (**A**) Body weight, (**B**) average daily gain, (**C**) average daily feed intake, and (**D**) feed/gain ratio of broiler chickens fed *B. velezensis* CML532. Jejunum gut morphometrics: (**E**) villus length, (**F**) crypt depth, and (**G**) villus length/crypt depth ratio. (**H**) Immune organ weight of broiler chickens fed *B. velezensis* CML532. The activity of (**I**) ALP, Na^+^/K^+^-ATPase, sucrase, maltase activities, and relative gene expression of (**J**) tight junction proteins and (**K**) nutrient transport-related genes in the jejunum and (**L**) relative gene expression of cytokines in the cecal tonsils of broiler chickens fed diets containing *B. velezensis* CML532. (**M**) Effects of *B. velezensis* CML532 supplementation on serum antioxidant of broiler chickens. Data are expressed as the mean ± SEM (*n* = 6). * *p* < 0.05, ** *p* < 0.01, *** *p* < 0.001.

**Figure 5 microorganisms-12-00771-f005:**
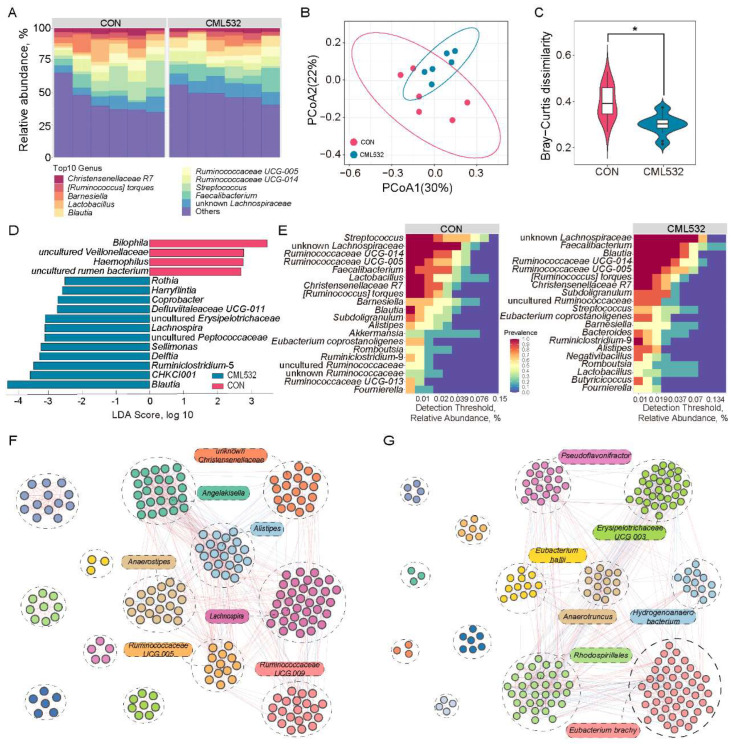
*B. velezensis* CML532 changes the structure of the cecal microbial community. (**A**) Relative abundance of major bacterial genus. (**B**,**C**) Principal coordinate analysis (PCoA) and Bray–Curtis dissimilarity of cecal microbiota composition. (**D**) The LEfSe analysis of the cecum microbiota at the genus level. (**E**) The difference in number of cores OTUs and their prevalence at different abundance thresholds (relative abundance). Highly connected modules within (**F**) CON and (**G**) *B. velezensis* CML532 networks. Red and blue links indicate positive and negative correlations, respectively. The boxes represent the more frequent co-occurrences (higher degree) of microbial taxa in the module. * *p* < 0.05.

**Figure 6 microorganisms-12-00771-f006:**
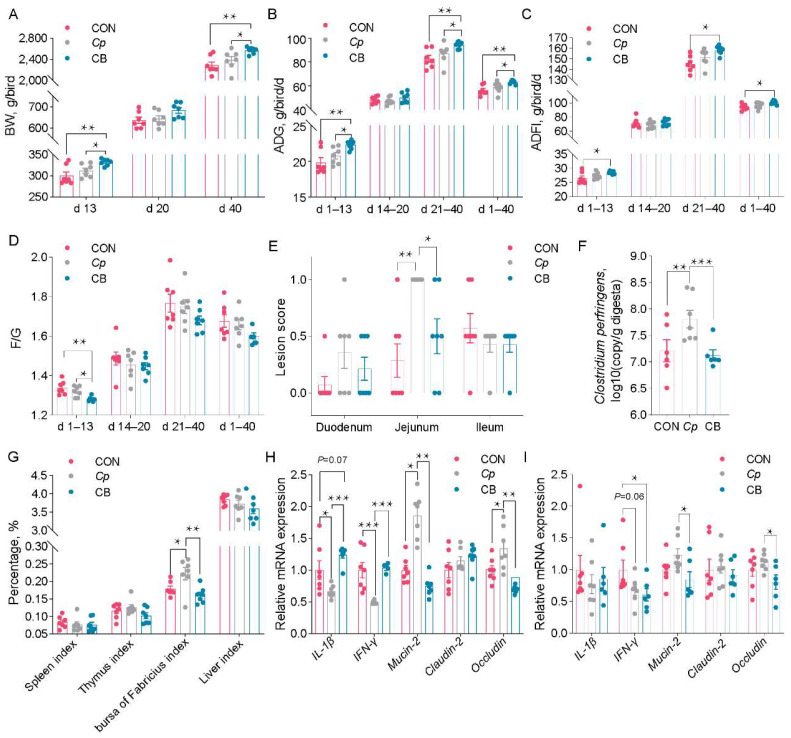
*B. velezensis* CML532 inhibits intestinal *C*. *perfringens* colonization and alleviated intestinal mucosal injury. (**A**) Body weight, (**B**) average daily gain, (**C**) average daily feed intake, and (**D**) feed/gain ratio of broiler chickens fed diets containing *B. velezensis* CML532 with *C. perfringens* challenge. (**E**) Intestinal lesion score of broilers. (**F**) The load of *C. perfringens* in the ileum. (**G**) Immune organ weight of broiler chickens fed *B. velezensis* CML532 with *C. perfringens* challenge. The relative gene expression of tight junction proteins and cytokines in the (**H**) jejunum and (**I**) ileum of broiler chickens fed *B. velezensis* CML532 with *C. perfringens* challenge. Data are expressed as the mean ± SEM (*n* = 7). * *p* < 0.05, ** *p* < 0.01, *** *p* < 0.001.

**Table 1 microorganisms-12-00771-t001:** Antibiotic susceptibility of *Bacillus velezensis* CML532.

Antibiotic	Sensitivity ^1^		Antibiotic	Sensitivity		Antibiotic	Sensitivity	
	F1 ^2^	F5 ^3^		F1	F5		F1	F5
Piperacillin/tazobactam	S	S	Meropenem	S	S	Rifampin	S	S
Ampicillin	S	S	Vancomycin	S	S	Tetracycline	R	R
Ciprofloxacin	S	S	Neomycin	S	S	Trimethoprim/ Sulfamethoxazole	S	S
Penicillin	S	S	Ceftriaxone	S	S	Amikacin	S	S
Erythromycin	S	S	Cephalosporin	S	S	Ceftazidime	S	S
Chloromycetin	S	S	Cefazolin	S	S	Cephalothin	S	S
Azithromycin	S	S	Cefotaxime	S	S	Cefoperazone	S	S
Clindamycin	S	S	Spectinomycin	S	S	Gentamicin	S	S
Doxycycline	S	S	Cefuroxime	S	S	Oxacillin	S	S
Clarithromycin	S	S	Minocycline	S	S	Nitrofurantoin	S	S

^1^ Resistant (R) or sensitive (S). ^2^ First-generation strain (F1); ^3^ fifth-generation strain (F5).

## Data Availability

The raw amplicon sequencing data from this study are available in the NCBI Sequence Read Archive (SRA) with the BioProject identifier PRJNA1062171. *Bacillus velezensis* CML532 strain has been deposited in the China General Microbiological Culture Collection Center (CGMCC, Beijing, China), accession number CGMCC 24752.

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
