# Peer review of "A New *Bacillus velezensis* Strain CML532 Improves Chicken Growth Performance and Reduces Intestinal *Clostridium perfringens* Colonization"

_microorganisms, 2024, doi:10.3390/microorganisms12040771_

Round 1

Reviewer 1 Report

Comments and Suggestions for Authors

1.-The authors describe in results diverse bacterial strains obtained from cecal samples of chicken from diverse regions of China (table 1). Although, the authors describe diverse species of Bacillus genus (lines 240-248), only results were described for the B. velezensis strains (line 250-270). So that, I suggest eliminate the table 1, or some explanation about the properties from other Bacillus species or they were not considered.

2.- From the results in section 3.1, the B. velezensis CML532 strain was selected. According with the results in figure 1D, What mean high levels of enzymatic activities?, also What mean stronger inhibitory effect against C. perfringens?. I suggest use some controls for these experiments.

3.- In results, section 3.3 and 3,5, comparing data of significant increase or decrease during the impacts of B. velezensis CML532 on growth performance is necessary, is difficult the interpretation of figure 4 and 6, without explanation in text. Also is similar for data of gene expression. A detailed description of results must be addressed for clarity.

Author Response

Dear Editor and reviewers:

Thank you very much for your comments and professional advice for our manuscript entitled “A New Bacillus velezensis Strain CML532 Improves Chicken Growth Performance and Reduces Intestinal Clostridium perfringens Colonization (microorganisms-2899699)”. We greatly appreciated you have made insightful comments and thoughtful guidance to our work which greatly improve academic rigor of our article. Accordingly, we have addressed all comments in the revised manuscript. We have revised the manuscript accordingly and highlighted the changes in yellow. Following are the answers to reviewers.

1.-The authors describe in results diverse bacterial strains obtained from cecal samples of chicken from diverse regions of China (table 1). Although, the authors describe diverse species of Bacillus genus (lines 240-248), only results were described for the B. velezensis strains (line 250-270). So that, I suggest eliminate the table 1, or some explanation about the properties from other Bacillus species or they were not considered.

AU: Thank you for your suggestion. In this study, we first isolated Bacillus strains from cecal samples of chicken, and then the Bacillus velezensis strains were screened from all the strains listed in Table 1. We think the results in Table 1 should be retained, while according to your comments, we removed Table 1 to the supplementary materials (Line 261).

2.- From the results in section 3.1, the B. velezensis CML532 strain was selected. According with the results in figure 1D, What mean high levels of enzymatic activities?, also What mean stronger inhibitory effect against C. perfringens?. I suggest use some controls for these experiments.

AU: We have revised the descriptions of this part of the results to avoid misunderstandings (Lines 283-287).

3.- In results, section 3.3 and 3,5, comparing data of significant increase or decrease during the impacts of B. velezensis CML532 on growth performance is necessary, is difficult the interpretation of figure 4 and 6, without explanation in text. Also is similar for data of gene expression. A detailed description of results must be addressed for clarity.

AU: According to your comments, we have provided an extensive description of the growth performance and gene expression data in sections 3.3 and 3.5 (Lines 342-345; 433-436).

Reviewer 2 Report

Comments and Suggestions for Authors

The manuscript presented to me for review contains a great deal of cognitively valuable results. It was prepared in an appropriate manner. At the same time, out of my duty as a reviewer, I would like to ask for clarification of a few issues posted below:

Line 92 - on what basis was the potential affiliation with Bacillus inferred?

If I understood correctly, the authors created two independent experiments at the same time. Is this the case? Please indicate the numbers of birds in both tests. Were the tests performed simultaneously? The timing of the body weight and feed conversion measurements were used rather unusually for both tests. Why? 

There is a lack of information on the maintenance conditions of the birds. 

At the same time, in the material and methods section, not all the reagents and equipment were labeled with the manufacturer's name. Please supplement this.

The description of statistical methods is unsatisfactory. Please indicate the normality of distribution test used. Which data were counted with a nonparametric test? Which parametric etst was used?

Figure 6 is very difficult to analyze. Please consider converting it to a table.

Author Response

Dear Editor and reviewers:

Thank you very much for your comments and professional advice for our manuscript entitled “A New Bacillus velezensis Strain CML532 Improves Chicken Growth Performance and Reduces Intestinal Clostridium perfringens Colonization (microorganisms-2899699)”. We greatly appreciated you have made insightful comments and thoughtful guidance to our work which greatly improve academic rigor of our article. Accordingly, we have addressed all comments in the revised manuscript. We have revised the manuscript accordingly and highlighted the changes in yellow. Following are the answers to reviewers.

The manuscript presented to me for review contains a great deal of cognitively valuable results. It was prepared in an appropriate manner. At the same time, out of my duty as a reviewer, I would like to ask for clarification of a few issues posted below:

AU: We appreciate these encouraging comments.

1.Line 92 - on what basis was the potential affiliation with Bacillus inferred?

AU: All the strains were identified through sequencing its 16S rRNA gene and BLAST against Database, as we described in the Materials and Methods (Lines 96-102).

2.If I understood correctly, the authors created two independent experiments at the same time. Is this the case? Please indicate the numbers of birds in both tests. Were the tests performed simultaneously? The timing of the body weight and feed conversion measurements were used rather unusually for both tests. Why?

AU: We have conducted two independent experiments; therefore, the timing of body weight and feed conversion measurements is variable. The numbers of birds in both experiments can be found in the Materials and Methods section (Lines 136-137, Lines 153-154). To be clear, we also added the number of birds used in each trial in the Results section (Line 341, Lines 416-418).

3.There is a lack of information on the maintenance conditions of the birds.

AU: We have added a section on bird management to the Materials and Methods section (Line 165-171).

4.At the same time, in the material and methods section, not all the reagents and equipment were labeled with the manufacturer's name. Please supplement this.

AU: We have added the manufacturer’s name of the reagents and equipment to the Materials and Methods section (Line 83-256).

5.The description of statistical methods is unsatisfactory. Please indicate the normality of distribution test used. Which data were counted with a nonparametric test? Which parametric etst was used?

AU: Revised as suggested (Line 249-255).

6.Figure 6 is very difficult to analyze. Please consider converting it to a table.

AU: Thank you for your suggestion. Figure 6 clearly exhibits the growth performance and related physiological and biochemical data of broilers after C. perfringens challenge, with statistically significant differences highlighted with stars. If shown in the format of Table, 13 tables are needed. So, we retained this Figure.